# Hyperspectral UAV Images at Different Altitudes for Monitoring the Leaf Nitrogen Content in Cotton Crops

**Caixia Yin [1], Xin Lv [1], Lifu Zhang [1,2], Lulu Ma [1], Huihan Wang [1], Linshan Zhang [2] and Ze Zhang [1,\*]**

[1] College of Agriculture, Shihezi University, Shihezi 832003, China; yincaixia@stu.shzu.edu.cn (C.Y.); luxin@shzu.edu.cn (X.L.); zhanglf@radi.ac.cn (L.Z.); malulu@stu.shzu.edu.cn (L.M.); 20192012028@stu.shzu.edu.cn (H.W.)

[2] Institute of Aerospace Information Innovation, Chinese Academy of Sciences, Beijing 100080, China; zhangls@aircas.ac.cn

[\*] Correspondence: zhangze1227@shzu.edu.cn; Tel.: +86-139-9953-4031

**Abstract:** The accurate assessment of cotton nitrogen (N) content over a large area using an unmanned aerial vehicle (UAV) and a hyperspectral meter has practical significance for the precise management of cotton N fertilizer. In this study, we tested the feasibility of the use of a UAV equipped with a hyperspectral spectrometer for monitoring cotton leaf nitrogen content (LNC) by analyzing spectral reflectance (SR) data collected by the UAV flying at altitudes of 60, 80, and 100 m. The experiments performed included two cotton varieties and six N treatments, with applications ranging from 0 to 480 kg ha$^{-1}$. The results showed the following: (i) With the increase in UAV flight altitude, SR at 500–550 nm increases. In the near-infrared range, SR decreases with the increase in UAV flight altitude. The unique characteristics of vegetation comprise a decrease in the "green peak", a "red valley" increase, and a redshift appearing in the "red edge" position. (ii) We completed the unsupervised classification of images and found that after classification, the SR was significantly correlated to the cotton LNC in both the visible and near-infrared regions. Before classification, the relationship between spectral data and LNC was not significant. (iii) Fusion modeling showed improved performance when UAV data were collected at three different heights. The model established by multiple linear regression (MLR) had the best performance of those tested in this study, where the model-adjusted the coefficient of determination ($R^2$), root-mean-square error (RMSE), and mean absolute error (MAE) reached 0.96, 1.12, and 1.57, respectively. This was followed by support vector regression (SVR), for which the adjusted_$R^2$, RMSE, and MAE reached 0.71, 1.48, and 1.08, respectively. The worst performance was found for principal component regression (PCR), for which the adjusted_$R^2$, RMSE, and MAE reached 0.59, 1.74, and 1.36, respectively. Therefore, we can conclude that taking UAV hyperspectral images at multiple heights results in a more comprehensive reflection of canopy information and, thus, has greater potential for monitoring cotton LNC.

**Keywords:** UAV; spectral reflectance (SR); leaf nitrogen content (LNC); cotton

## 1. Introduction

Nitrogen (N) is an essential element for all crops, and the accurate monitoring of N is vital to improve crop yields and protect the environment [1]. The improper use of N is currently one of the most serious environmental problems [2]. Monitoring the N nutrition of cotton can provide accurate information about N application, reduce waste from excessive N fertilizer application, reduce labor costs, and improve the cotton yield, thus resulting in planting benefits [3,4]. In recent years, studies have increasingly reported on the use of spectral remote sensing technology in crop nutrition monitoring [5,6]. They have shown that chlorophyll can absorb 70–90% of the red and blue light in incident light and emit a large amount of green and near-infrared light [7]. Leaf chlorophyll has a strong linear relationship with leaf nitrogen concentration ($r^2 = 0.83$) [8]. Hyperspectral sensors

measure reflected energy in narrower bands and thus measure more information due to the detection of a wider range of spectral bands [9].

The overlap of absorption spectra for different photosynthetic pigments in leaves is regarded as an important limitation when estimating nitrogen content based on a single narrow-band reflection [10]. When filtering out excess information, the spectral index is a reliable variable for estimating nitrogen content. Some researchers have developed specific vegetation indexes based on reflectance to estimate plant nitrogen content [11,12], such as the best normalized difference index (NDIopt) [1], bimodal canopy nitrogen index (DCNI) [13], normalized difference red edge (NDRE) [14], and pixel best index (PBI) [10]. Additionally, the spectral index has been widely used in crop nutrition monitoring [1,12,13,15–18], and the monitoring accuracy has reached a high level [19,20]. However, the stability of the index is limited by many factors, such as crop growth phenology, soil exposure, and crop growth status, among others, which have a negative impact on the usability of the index. In addition, the spatial resolution will affect the calculation of the vegetation index. Some researchers proposed that obtaining spectral data from different heights might result in more significantly abundant information [21,22].

UAV vegetation indexes (Vis) contribute more information to model algorithms than single spectral band information [23]; however, these indexes can easily become saturated under medium or high vegetation coverage [24]. One study showed that a UAV nitrogen monitoring model of wheat performed well in the early stage of crop growth [25]. Based on the leaf spectrum or near-ground canopy spectrum, the status of N could only be measured for a single sample point at a time. Near-ground spectrum monitoring technology is suitable for information monitoring of small-area crops [26,27]. UAV hyperspectral imaging expands the measurement range from sample points to the regional scale, which can provide more practical monitoring results [28]. With few restrictions other than working in harsh weather conditions, UAV-based remote sensing technology has many potential applications in precision agriculture [29–33] and has been widely used to monitor nitrogen content [34,35].

Although the accuracy of nitrogen monitoring based on UAV hyperspectral remote sensing has made great progress, it is difficult to achieve satisfactory performance with regard to the model application accuracy. In addition to ensuring the stability of the model in a changeable environment, the choice of index and model may be significantly adapted to the complex environment of farmland. The traditional linear regression method has defects in model fitting, and the model accuracy is often low [36]. With the development of machine learning technology, algorithms such as random forest (RF), partial least-squares regression (PLSR), and support vector regression (SVR) are widely used in the inversion of crop biochemical parameters and have gradually replaced the traditional stepwise regression method due to their better performance. For example, PLSR combines the advantages of multiple linear regression (MLR), principal component regression (PCR), and principal component analysis (PCA), and has superior interpretability and prediction functionality. SVR can combine high-dimensional data and noise modes to improve prediction accuracy. RF more rapidly makes a decision that is more robust through the bootstrap sampling of training data and random selection of variables [37–39]. Partial least-squares regression (PLSR) overcomes the collinearity and overfitting problems found in multiple linear regression (MLR) while using as much information as possible [40,41]. The combination of the logarithm of reflectance and PLS may have a better effect on nitrogen monitoring [40]. The random forest method is more suitable for predicting nitrogen accumulation in maize. Machine learning has been widely used to train models [40,42–44].

Therefore, in this study, we flew a UAV at different altitudes to construct the leaf nitrogen content (LNC) monitoring model. The purpose of using different flight heights was to obtain more all-round information regarding the cotton canopy. At the same time, in order to reduce the impact of mixed pixels on nitrogen monitoring accuracy, we adopted an unsupervised classification of images and selected image pixels with relatively little interference, with the subsequent calculation of the vegetation index. Finally, we

constructed a nitrogen content monitoring model by taking the three altitudes obtained from the UAV vegetation indexes as independent variables. The model thus integrates more canopy information for cotton and is expected to be relatively more stable.

## 2. Materials and Methods

This study was conducted in Shihezi (44°19′ N, 85°59′ E) in Xinjiang, China. The experimental area is shown in Figure 1. The average annual temperature is between 6.5 and 7.2 °C. The highest annual temperature occurs in July, with an average of 25.1–26.1 °C; the lowest temperature occurs in the south of the area in January, with an average of −18.6–15.5 °C. The annual precipitation is between 125.0 and 207.7 mm. Sunshine is abundant and the area has a typical temperate continental climate [45].

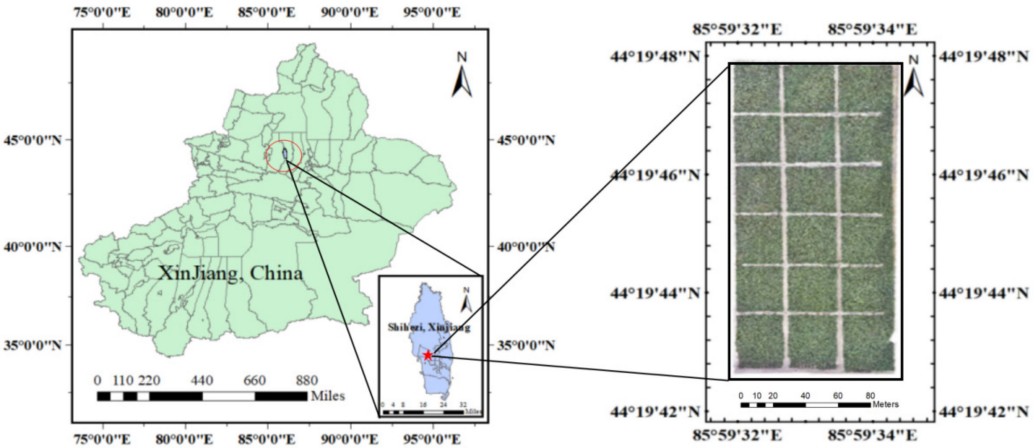

**Figure 1.** The experimental area.

This cultivated land has been planted with cotton for more than three years. Thirty percent of the total fertilizer was applied before cotton was planted to follow the objective law of locally applied fertilization and ensure the normal growth of cotton, regardless of its economic benefits. After seedling emergence, 70% N fertilizer was applied on 22 June, 9 July, 18 July, 5 August, and 15 August at the proportions of 10%, 10%, 20%, 20%, and 10%, respectively, to create a linear N gradient. The water required was measured by local standard drip irrigation on 13 June, 23 June, 14 July, 25 July, 5 August, and 16 August. A total of 6 different fertilizer rates were set in the experiment. The fertilizer rates were 0, 120, 240, 330, 360, and 480 kg/ha, which were labeled as $N_0$, $N_{120}$, $N_{240}$, $N_{330}$, $N_{360}$, and $N_{480}$, respectively. Among them, 330 kg ha$^{-1}$ is the conventional fertilization amount for local cotton (Table 1).

**Table 1.** Urea (46%-N) application status (taking 300 kg ha$^{-1}$ as an example).

| Base Fertilizer | N Application Stage | | |
|---|---|---|---|
| | Data | Proportion | Detailed Fertilizer Usage (kg ha$^{-1}$) |
| 30% | 70% (330 × 0.7 = 231 kg/ha) | | |
| 99 kg/ha | 13 June | 12% | 27.72 |
| | 23 June | 12% | 27.72 |
| | 14 July | 15% | 34.65 |
| | 25 July | 17% | 39.27 |
| | 5 August | 20% | 46.2 |
| | 16 August | 24% | 55.44 |

### 2.1. Experiment Management and Data Extraction

#### 2.1.1. UAV Platforms and Airborne Sensors

We used a UAV platform equipped with airborne hyperspectral sensors. The UAV was a DJI M600 series aircraft (Figure 2), which was made in China with a maximum altitude of 2500 m and a maximum horizontal flight speed of 65 km/h in a windless environment. The UAV weighed 6 kg and had a maximum ascent speed of 5 m/s and a maximum descent speed of 3 m/s. When the UAV was flying, the course and side overlap rate was set at 75%, and the corresponding ground resolution was approximately 2.82 cm. The sensor was named Nano-Hyperspec (Figure 3) and weighed 0.6 kg and had 640 space passages. The sampling interval was 2.2 nm. The number of spectral channels was 270, and the band range was 400–1000 nm. A whiteboard was placed in the middle of the flight area to facilitate the joint acquisition of the whiteboard and ground objects during flight.

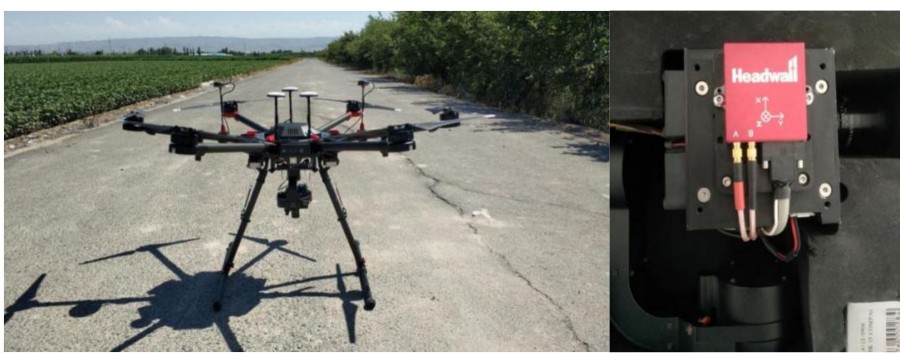

**Figure 2.** DJI Matrice 600 series aircraft equipped with the GPS and a sensor (**left**). The Nano-Hyperspec sensor (**right**).

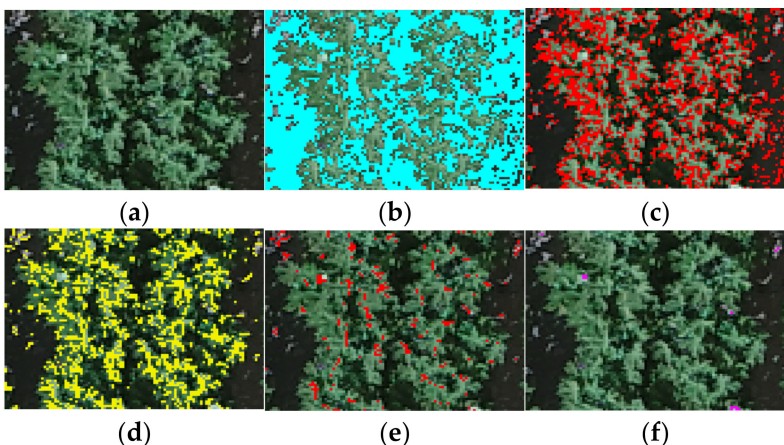

**Figure 3.** Image extraction range. (**a**) is a part of the image intercepted from the canopy RGB image, (**b**) refers to irrelevant substances such as soil and plastic film, (**c**–**f**) refers to various leaf postures of the photographed cotton canopy. It should be noted that there are only 5 categories here, and 49 categories will be used to show the posture of cotton leaves in more detail in this study.

The UAV flew at three altitudes of 60, 80, and 100 m away from the cotton canopy. When the UAV flew at a 100 m altitude, the ground spatial resolution was 0.06 m. Image correction was divided into orthorectification and radiometric calibration. The sensor was equipped with a GPS to facilitate orthophoto correction. Radiometric calibration was divided into atmospheric correction and radiometric correction. Radiation correction was performed using color plates with a reflectance of 100%. The principle of radiation correction is shown in Equation (1). W_DN, W_Ref, V_DN, and V_Ref refer to the digital number of the whiteboard, whiteboard reflectance, the digital number of the vegetation,

and vegetation reflectance, respectively. The ENVI Classic 5.3 was used for UAV image mosaicing.

$$W\_DN/W\_Ref = V\_DN/V\_Ref \tag{1}$$

2.1.2. Hyperspectral Reflectance Extraction of UAV

According to the longitude and latitude coordinates of the ground acquisition sample points, feature points with the same name were found on the image. In this study, we extracted spectral information at the image position corresponding to the ground sampling point, referred to as the region of interest (ROI). For image classification, we adopted an unsupervised classification method. Due to there being a number of cotton canopy leaves and different leaf states, the obtained image pixels mostly comprised mixed information. In this study, we classified those with similar pixel mixing information into category 1. We assumed that there were 49 categories, which were regarded as 49 specific states of the cotton canopy, and these were visually displayed (Figure 3: the cotton canopy is simply divided into five categories for viewing; Figure 4: RGB images and spectral curves corresponding to several pixels). The spectral data corresponding to each category of pixels were obtained at the image position corresponding to the ground sampling point. For the obtained spectral data, these methods were used to process the reflectance: SG smoothing, Gaussian filter, and standard normal variate (SNV).

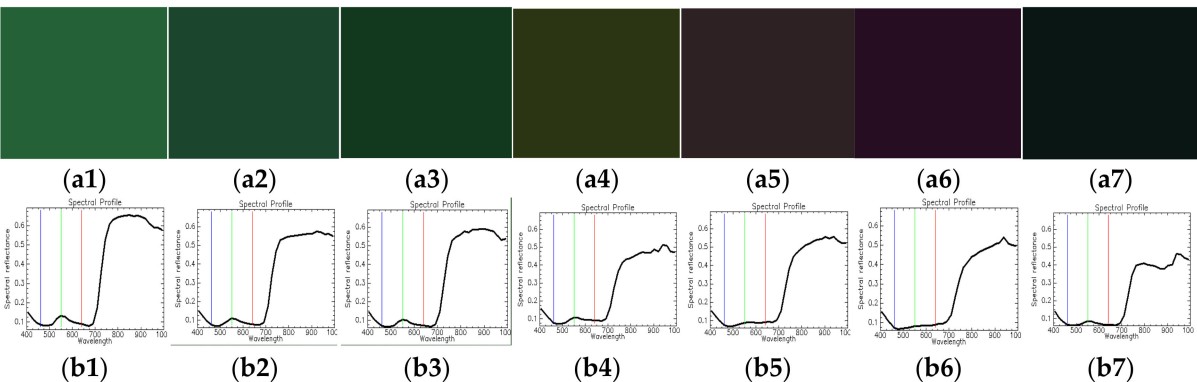

**Figure 4.** (**a1**–**a7**) refers to the pixel display diagram in which several typical pixels are randomly selected and combined with red light, green light, and blue light. (**b1**–**b7**) are corresponding spectral curves (the red, green, and blue lines in figures (**b1**–**b7**) represent the positions of red light, green light, and blue light, respectively). The typical pixel spectrum after image classification is shown as an example. With the deepening of the pixel RGB map (**a1**–**a7**), the spectral curve has a visible decline in the near-infrared region (**b1**–**b7**), indicating that the mixed pixels are seriously affected by the soil.

*2.2. Collection of Leaf Samples and Laboratory Measurement of the Leaf Nitrogen Content (LNC)*

LNC: During the blooming period (17 July), we harvested cotton plants, excised the leaves, and took them back to the laboratory to measure the LNC. The samples were then oven-dried at 105 °C for 30 min, followed by drying at 80 °C until constant weights were reached. Leaf nitrogen content (LNC) was determined by micro Kjeldahl analysis [46].

*2.3. Spectral Pretreatment Method*

Original SR: The original spectral reflectance (SR) refers to the reflectance directly obtained by the UAV spectrometer without any form of transformation and processing.

SG smoothness: The basic idea of spectral smoothing is to take several points before and after the smoothing points for averaging or fitting to obtain the best-estimated value of the smoothing points and eliminate random noise. In this study, the smoothing denoising method proposed by Savitzky and Golay referred to as SG smoothing was used for spectral smoothing. The most significant feature of this method is that it can filter noise while

maintaining the shape and width of the signal. The number of window points in this study was 11.

Gaussian filter: A Gaussian filter is a kind of linear smoothing filter that is suitable for eliminating Gaussian noise and is widely used in noise reduction during image processing. Generally speaking, the weighted average of the whole image is determined by Gaussian filtering. The value of each pixel is obtained by the weighted average of that pixel and other pixel values in the neighborhood. The specific operation of Gaussian filtering involves scanning each pixel in the image with a template (or convolution mask) and replacing the value of the central pixel of the template with the weighted average gray value of the pixels in the neighborhood determined by the template [47].

Standard normal variate (SNV): The standard normal variate method involves standardizing the original normal data or normal variables. According to the linear consistency of normal distribution, data are transformed into standard normal distribution. Through standardized transformation, the values of different random variables can be easily compared, and this is convenient for practical research.

### 2.4. Build and Validate Models

In this paper, we focus on the linear single correlation coefficient (r), which is shown in Equation (2). The higher the correlation coefficient, the stronger the corresponding relationship between the evaluated components. $x_i$ and $y_i$ represent components 1 and 2, respectively. $\bar{x}$ and $\bar{y}$ represent the average of components 1 and 2, respectively.

$$r = \frac{\sum_{i=1}^{n}(x_i - \bar{x})(y_i - \bar{y})}{\sqrt{\sum_{i=1}^{n}(x_i - \bar{x})^2} \times \sqrt{\sum_{i=1}^{n}(y_i - \bar{y})^2}} \tag{2}$$

① Multiple linear regression, MLR.

Multiple stepwise linear-regression-generated equations relate canopy vegetation index (VI) properties to LNC. The related parameters in this paper are shown in Table 2. As independent variable sets, the VI variables are independent of each other.

**Table 2.** Relevant parameters used in multiple linear regression (MLR) in this study.

| Parameter | The Value |
| --- | --- |
| Leverage limit | 2.0 |
| Sample outlier limit, calibration | 3.0 |
| Individual value outlier, calibration | 3.0 |
| Individual value outlier, validation | 3.0 |
| Variable outlier limit, calibration | 2.0 |
| Variable outlier limit, validation | 3.0 |
| Total explained variance (%) | 20 |
| Ratio of calibrated to validated residual variance | 0.5 |
| Ratio of validated to calibrated residual variance | 0.70 |
| Residual variance increase limit (%) | 6.0 |

② Partial least-squares regression, PLSR.

PLSR can eliminate the multicollinearity (correlation) between independent variables, extract the most explanatory comprehensive variables from multiple variables, and establish an accurate and stable model. Due to the high correlation between adjacent bands of hyperspectral data, PLSR was suitable for analysis and modeling based on hyperspectral data. The model input algorithm used in this study was kernel PLS; the ratio of calibrated to validated residual variance was 0.5. The ratio of validated to calibrated residual variance was 0.75. The residual variance increase limit was 6.0%.

③ Principal component regression, PCR.

PCR uses the idea of dimensionality reduction to reduce the number of independent variables with little loss of information. Fewer independent variables are usually needed, so their role is self-evident. The model input algorithm used in this study was single value decomposition (SVD). The parameters of the model warnings are consistent with PLSR.

④ Support vector regression, SVR.

SVR is a machine learning algorithm based on statistical theory in which the input vector data are mapped from the original space to the high-dimensional space (HS) through nonlinear or linear mapping, and the optimal regression function is constructed in HS. SVR has advantages in solving nonlinear problems where the number of samples is less than the number of variables, so it has been widely used in agricultural quantitative remote sensing. The SVM type in this study was a regression (epsilon SVR). The kernel type was sigmoid; gamma, epsilon value, and cross-validation segment size were 0.01190476, 0.1, and 10, respectively.

### 2.5. Evaluation of Model Performance

In this study, the model accuracy was evaluated by three representative indicators: coefficient of determination ($R^2$), root-mean-square error (RMSE), and mean absolute error (MAE). The calculation formulas for each accuracy pointer are as follows. $\hat{y_i}$ refers to the predicted value and $y_i$ refers to the measured value. $R^2$ is the test result of the fitting degree (correlation) between the measured value and the predicted data, RMSE is the deviation degree of the predicted value compared with the measured value, and RMSE has the characteristic trend of increasing with the increase in the average value of the data. When the $R^2$ value is larger and the RMSE value is smaller, then the predicted value is closer to the measured value, indicating the prediction accuracy of the model is higher and thus has a better prediction effect. MAE indicates the degree of the average value of the model error, which can evaluate well the proximity between the measured value and the predicted value. When the MAE value is lower, the prediction accuracy of the model is higher and the prediction ability is stronger.

$$R2 = \frac{\sum_{i=1}^{n}(\hat{y_i} - \bar{y})^2}{\sum_{i=1}^{n}(y_i - \bar{y})^2} \tag{3}$$

$$RMSE = \sqrt{\frac{1}{n}\sum_{i=1}^{n}(\hat{y_i} - y_i)^2} \tag{4}$$

$$MAE = \frac{1}{n}\sum_{i=1}^{n}|\hat{y_i} - y_i| \tag{5}$$

### 2.6. The Software

ENVI Classic 5.3 was used to obtain and process the SR from the UAV images; IBM SPSS Statistics 24 was used for statistical analysis. Unscrambler X 10.4 was used for spectrum processing and modeling and Origin 2018 was used for taking pictures.

## 3. Results

### 3.1. The Regular SR Multiple Scales

The correlation coefficients of SR between different bands (Figure 5) were analyzed. The correlation coefficient between bands of SR at the same height was relatively high, though was higher between the red edge and near-infrared region, and the lowest value was approximately 0.8. We know that a lot of duplicated information is found in hyperspectral data. We chose the appropriate bands to monitor leaf nitrogen content (LNC) from hyperspectral data, which may provide opportunities for the use of multi-spectral data and reduce economic costs to facilitate acceptance by farmers. The correlation of the SR

at different altitudes (between 60 and 80 m, 80 and 100 m, and 60 and 100 m UAV height) was analyzed (Figure 5). In visible light, the lowest coefficient was approximately 0.2. Overall, the reflectance correlation among visible light, red edge, and near-infrared was less than 0.4.

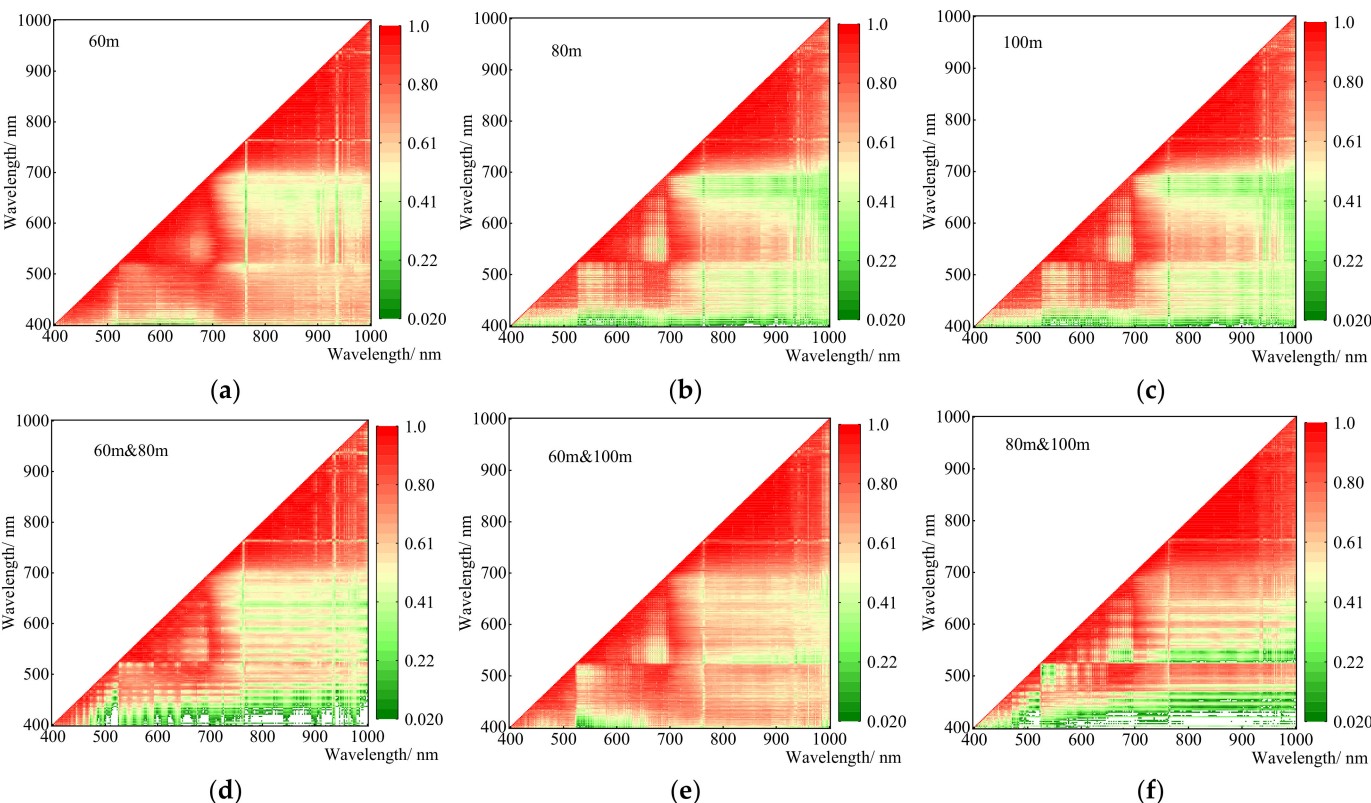

**Figure 5.** Correlation coefficient between spectral reflectance (SR) and SR at the same heights (**a–c**) and correlation coefficient between SR and SR at different heights (**d–f**).

Taking specific vegetation pixels as an example, the SR of vegetation varied significantly at different heights (Figure 6). With the increase in height, the unique features of vegetation resulted in a decrease in the "green peak" (vegetation has a reflection peak near 540 nm), an increase in the "red valley" (vegetation has an absorption peak near 660 nm), and a "red edge" position (the reflectance of green vegetation increases rapidly between 670 and 760 nm) that appeared as a redshift. The SR at 500–550 nm increased with the flight height of the UAV. In the near-infrared range, the SR decreased with increasing UAV flying height. This is because the image pixels obtained for cotton leaves at a 60 m flight altitude were clearer than those with SR at 100 m. The pixel spectrum is more affected by soil. Additionally, the near-infrared reflection curve rises sharply. The correlations between 60 and 80 m, 60 and 100 m, and 80 and 100 m were analyzed (Figure 7). The SR data of 60 and 80 m, 60 and 100 m, and 80 and 100 m were highly correlated, especially in the visible range. It may be that the information expressed by visible light is simple and visible to the naked eye and not susceptible to other external factors. There was little correlation in the near-infrared region. As shown in Figure 7, regarding the correlations between the SR and LNC in cotton, we found that regardless of the altitude at which the UAV acquired data, the correlation between the original SR and LNC was not ideal.

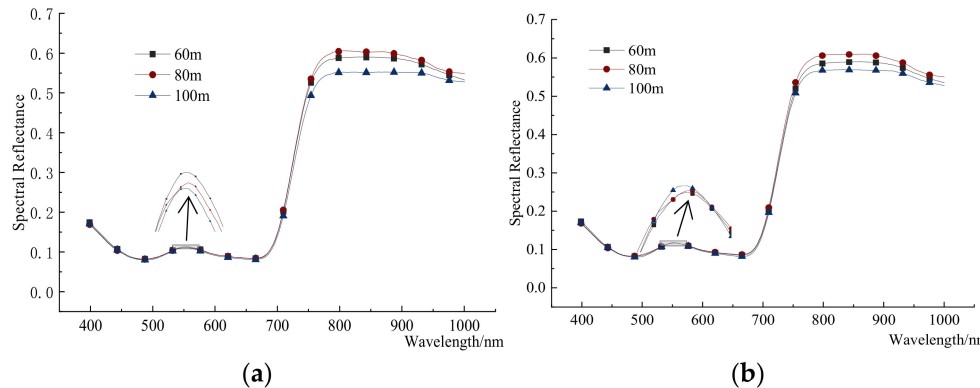

**Figure 6.** Change trend of hyperspectral reflectance of UAV at different heights (**a**,**b**) denote the test set and verification set, respectively, and the results are consistent).

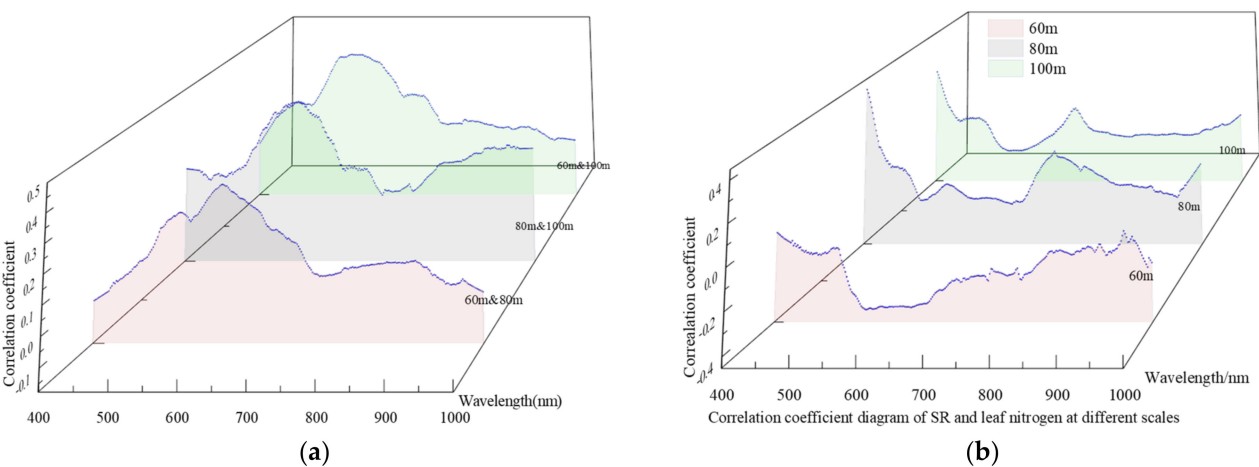

**Figure 7.** Correlation coefficient between hyperspectral reflectance at various heights (**a**) and correlation coefficient between SR and leaf nitrogen content (LNC) (**b**).

### 3.2. Research on Anti-Jamming Algorithm of UAV Hyperspectral Image

The cotton canopy images for three heights obtained on July 16 were selected to classify the images using unsupervised classification. It was assumed that each kind of pixel represented a specific state of the cotton canopy. The pixels were divided into 49 categories, which meant that the cotton canopy was regarded as 49 specific states. We extracted the SR in each specific state, and its mean value was used as the reflection information in this state. We selected one category that had a significant relationship with LNC and compared it with the spectral reflection information before classification, as shown in Figure 8. The results show that the classified pixels had significant differences, especially in the near-infrared region. Hence, we performed a simple Pearson correlation analysis between the SR, VI before and after classification, and LNC. From Figure 9 we can see that the SR after classification was significantly related to the LNC in both visible and near-infrared regions. The relationship between SR before classification and LNC was not significant. From Figure 10, we can see that the correlation coefficient between the classified spectral index and LNC was significantly improved regardless of the height. At 60 m, the correlation coefficients of VI 3, VI 7, VI 12, VI 13, and VI 22 reached more than 0.7.

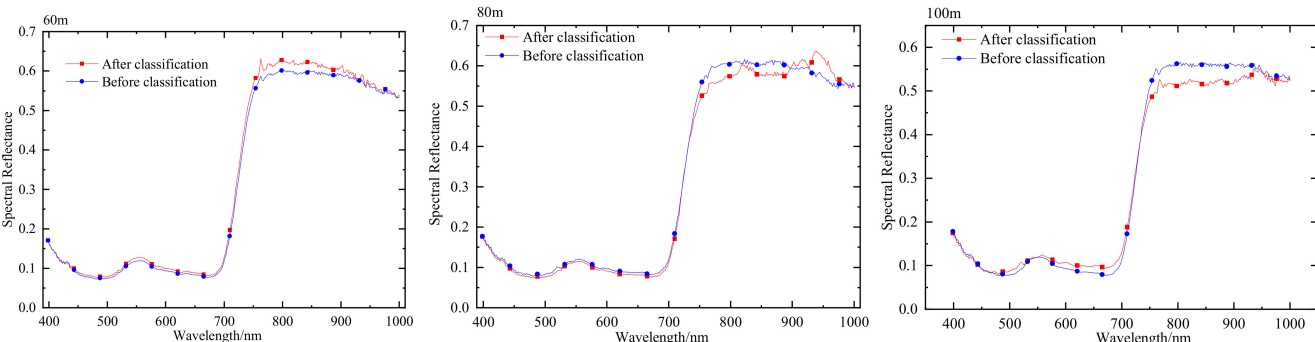

**Figure 8.** Hyperspectral reflectance before and after pixel classification of the UAV image.

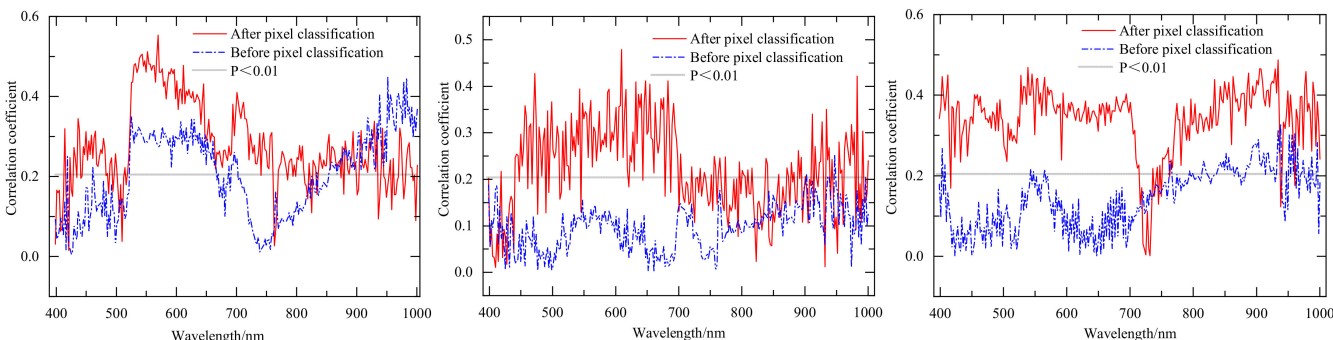

**Figure 9.** Correlation coefficient (absolute value) between SR and LNC in cotton before and after pixel classification.

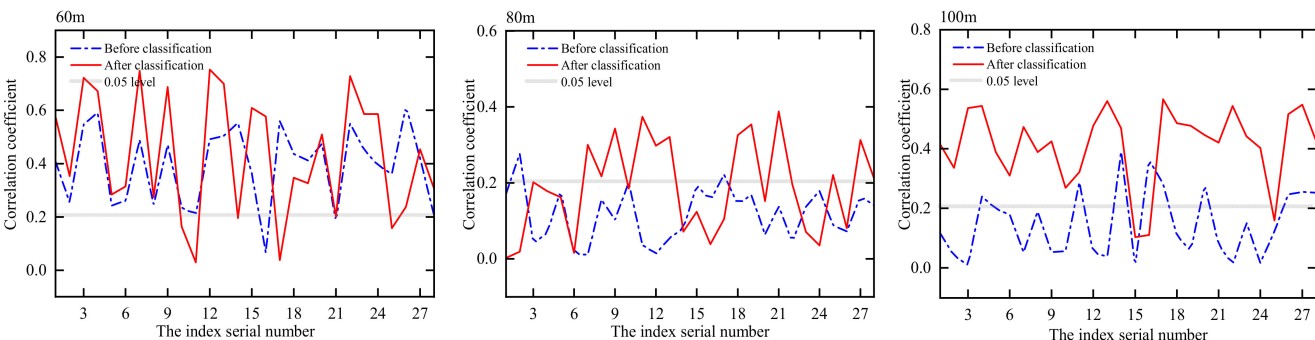

**Figure 10.** Correlation coefficient (absolute value) between vegetation index (VI) and LNC in cotton before and after pixel classification (the index was selected from another paper published by the author, which has a good nitrogen response relationship proposed by predecessors, DOI:10.1007/s12524-021-01355-0).

According to the experimental design, 108 data samples were obtained and the corresponding 28 related vegetation indexes were calculated. Through the error statistics and mean value calculation of the index, drawing and comparing the results (Figures 11–13), it can be seen from the figure that the error between spectral samples was significantly reduced after the classification of the reflectivity data obtained by the UAV at different flight altitudes. In particular, the sample error was significantly reduced when the image was classified. For the spectrum after SG smoothing and Gaussian filtering, the sample errors before and after classification were not very different. It can be seen that SG smoothing and Gaussian filtering can eliminate noise in the spectral data, which makes the spectral data stable. Pixel classification has a similar function, which makes the obtained spectral data less affected by soil and so on.

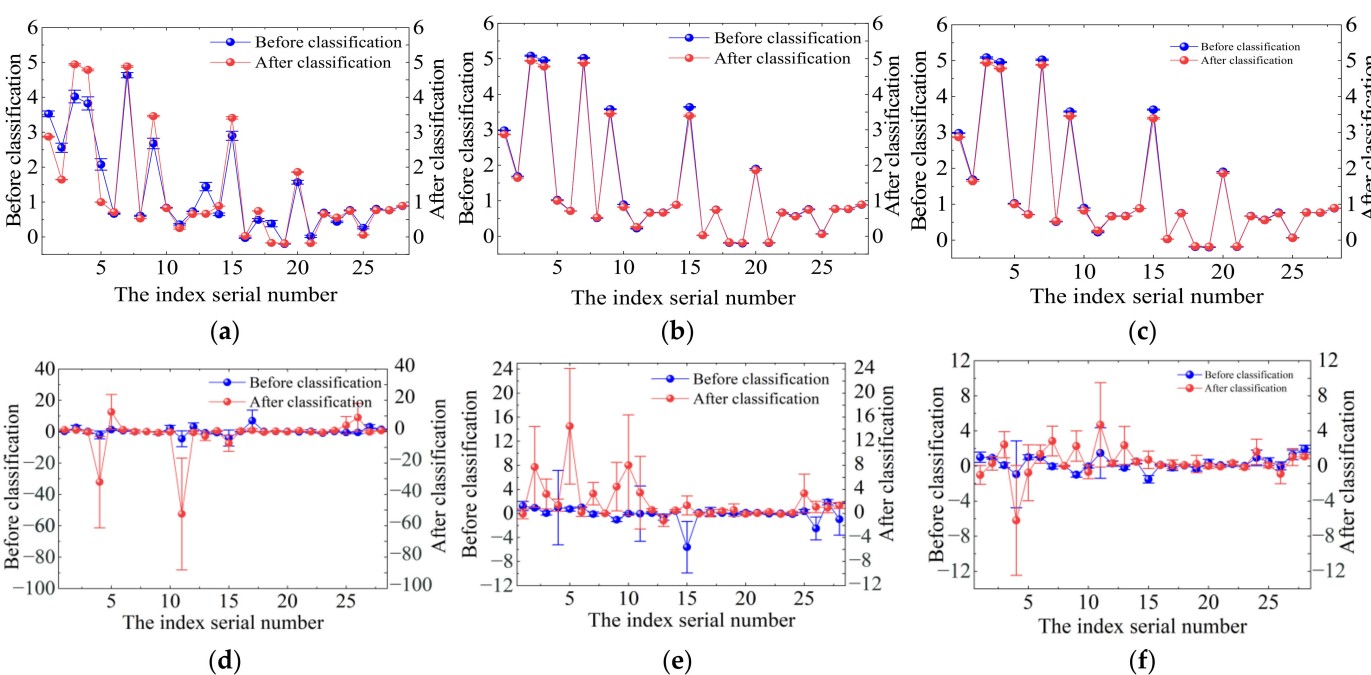

**Figure 11.** Change trend of some indexes before and after image pixel classification (60 m): (**a**) the original spectrum and after Gaussian filtering (**b**), after Gaussian filtering and then SG filtering (**c**), using the standard normal variate (SNV) method (**d**), after Gaussian and then SNV (**e**), and after successive Gaussian filtering, SG filtering, and SNV (**f**). The labels refer to the same for processing in Figures 12 and 13 for heights of 80 and 100 m, respectively.

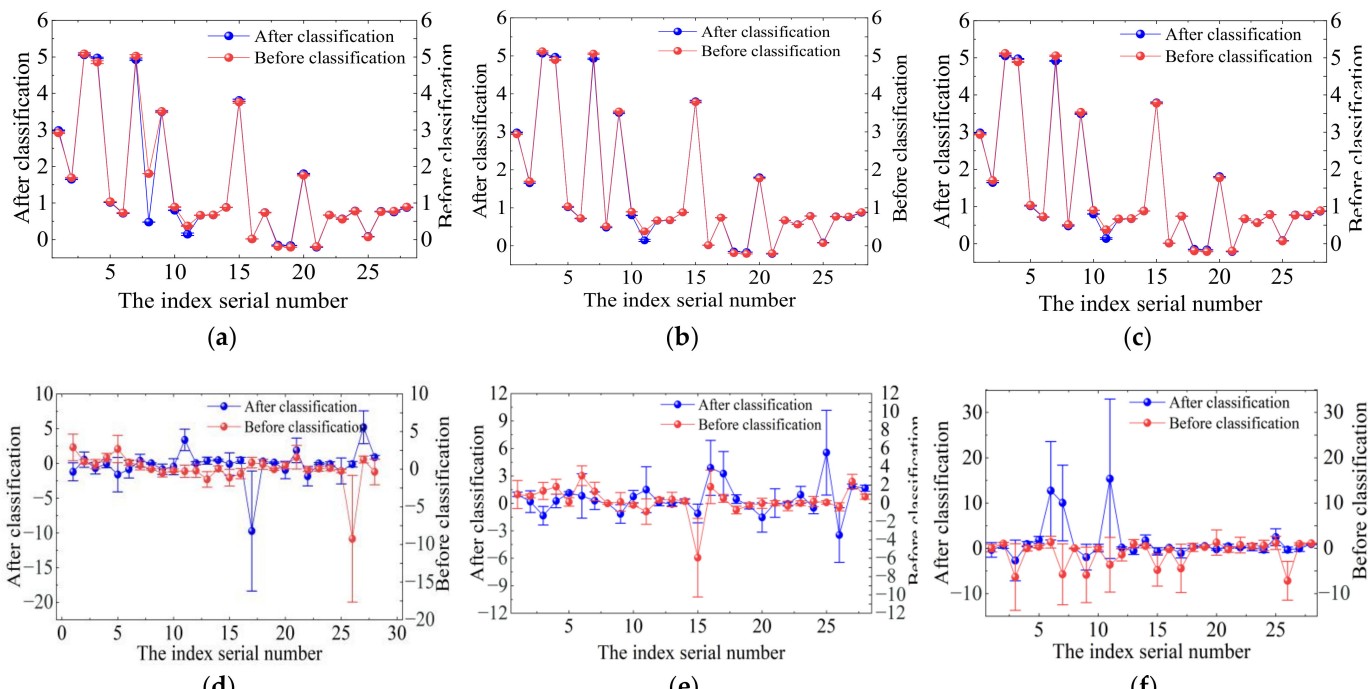

**Figure 12.** Change trend of some indexes before and after image pixel classification (80 m). Refer to Figure 11 for details of processing (**a**–**f**).

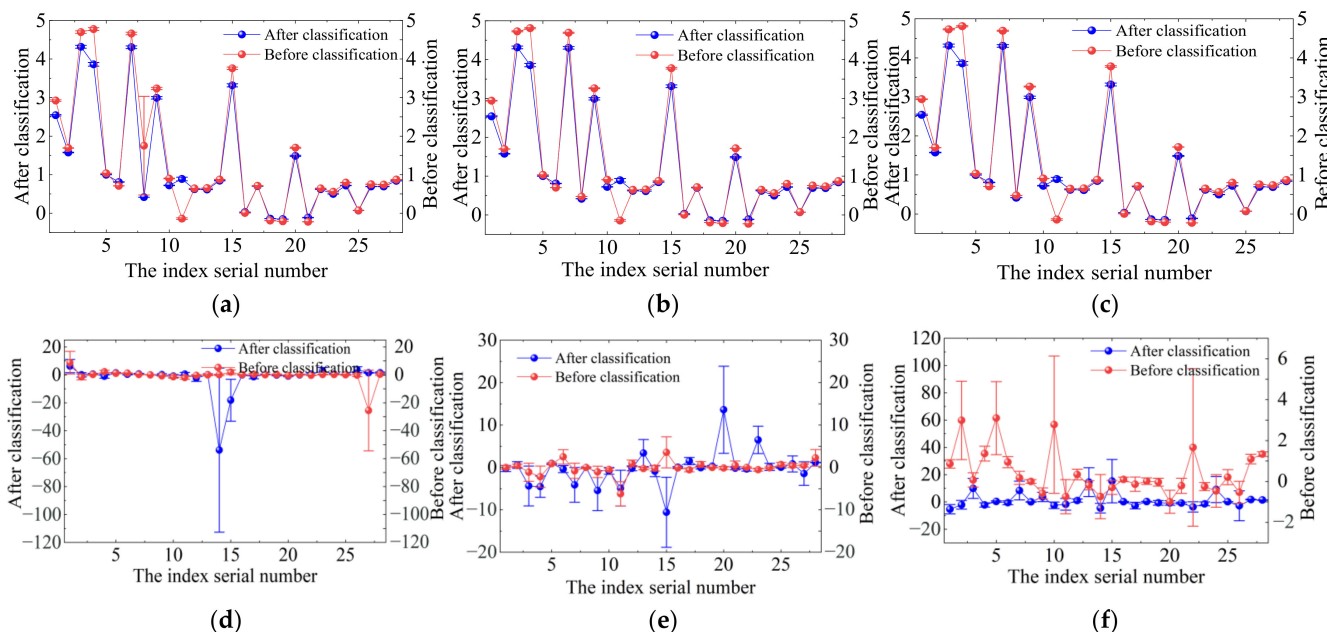

**Figure 13.** Change trend of some indexes before and after image pixel classification (100 m). Refer to Figure 11 for details of processing (**a**–**f**).

There were great differences in indexes 1, 2, 3, 4, 7, 9, 10, 11, and 15, and there was no significant difference in the other indexes after classification. Therefore, we can see that those indexes have poor stability and the others have relatively high stability. However, after SNV treatment, the values of indexes 2, 4, 5, 10, 11, 12, 13, 15, 17, 25, 26, and 27 changed greatly, while the others were not significantly different from before. It could be seen that different pretreatment methods play different roles for the index. After Gaussian filtering, the stability of the index obtained by the combination of SG smoothing and SNV was poor. Some indexes, such as 21–23, were relatively stable. It can be judged that these indexes have high stability and fewer interference factors. The VI corresponded to an 80 m flight altitude of the UAV; the original spectrum, Gaussian and SG smooth spectrum, and index were relatively stable. However, all indexes showed significant changes after SNV treatment. At 100 m, the indexes after SNV treatment had high stability except for indexes 14, 15, and 28. It can be seen that when the UAV flight altitude was relatively low, Gaussian smoothing and SG filtering were more suitable for image preprocessing, but SNV is recommended for image preprocessing as the UAV altitude increases.

We compared the exponential variation law under different pretreatment methods. At 60 m, the spectral sample error after SNV treatment was significantly increased. Comparing the pixel spectra before and after classification, it could be seen that the spectral index had various changes, and the index change degree was different. The response degree of different indexes to pixel classification was very different. After classification, spectral indexes 8 and 11 showed greater change than the other indexes, for which the bands used were 780, 700, 650, and 550 nm, respectively. The classification of visible pixels had a great influence on the corresponding values of these wavelengths.

### 3.3. The Model for Cotton LNC

We chose the VIs as the model variable with which to build models. Comparing the modeling results of different modeling methods (Table 3 and Figures 14–17), it could be concluded that the effects of different modeling methods on monitoring LNC varied greatly. This study shows that in the study of monitoring cotton LNC based on the UAV hyperspectral index, the MLR had a better effect. This was followed by PLS. SVR was slightly worse, and the effect of PCR was the worst. However, with the increase in UAV height, the modeling effect of SVR became worse. After data fusion modeling at three

heights (60, 80, and 100 m), it was found that the MLR performed better, but the verification effect was not ideal. Other modeling methods had little difference between the modeling set and the verification set, which showed that the other three methods had a relatively stable performance. The effect of SVR using a single height showed the worst performance, but the performance of the modeling effect was better after the fusion of three heights. Therefore, we can infer that SVR requires a large amount of modeling data.

**Table 3.** The final results of modeling.

| UAV Flight Altitude | Modeling Method | Model Performance (Test Set) | | | Model Performance (Validation Set) | | |
|---|---|---|---|---|---|---|---|
| | | $R^2$ | RMSE | MAE | $R^2$ | RMSE | MAE |
| 60 m | MLR | 0.80 | 0.41 | 0.95 | 0.63 | 1.66 | 1.31 |
| | PLS | 0.71 | 1.43 | 1.30 | 0.61 | 1.74 | 1.30 |
| | SVR | 0.67 | 1.77 | 1.17 | 0.62 | 1.68 | 1.31 |
| | PCR | 0.59 | 1.74 | 1.39 | 1.56 | 1.82 | 1.46 |
| 80 m | MLR | 0.72 | 1.67 | 1.16 | 1.47 | 1.99 | 1.61 |
| | PLSR | 0.49 | 1.92 | 1.56 | 0.35 | 2.19 | 1.79 |
| | SVR | 0.44 | 2.09 | 1.63 | 0.27 | 2.33 | 1.77 |
| | PCR | 0.26 | 2.34 | 1.89 | 0.19 | 2.47 | 2.00 |
| 100 m | MLR | 0.69 | 1.75 | 1.18 | 0.46 | 2.06 | 1.62 |
| | PLS | 0.61 | 1.69 | 1.35 | 0.47 | 1.97 | 1.56 |
| | PCR | 0.45 | 2.02 | 1.57 | 0.41 | 2.16 | 1.66 |
| | SVR | 0.40 | 1.67 | 1.59 | 0.29 | 2.31 | 1.16 |
| 60, 80, and 100 m | MLR | 0.96 | 1.12 | 1.57 | 0.47 | 2.43 | 1.57 |
| | SVR | 0.71 | 1.48 | 1.08 | 0.66 | 1.59 | 1.19 |
| | PLS | 0.63 | 1.66 | 1.19 | 0.58 | 1.77 | 1.36 |
| | PCR | 0.59 | 1.74 | 1.36 | 0.54 | 1.86 | 1.45 |

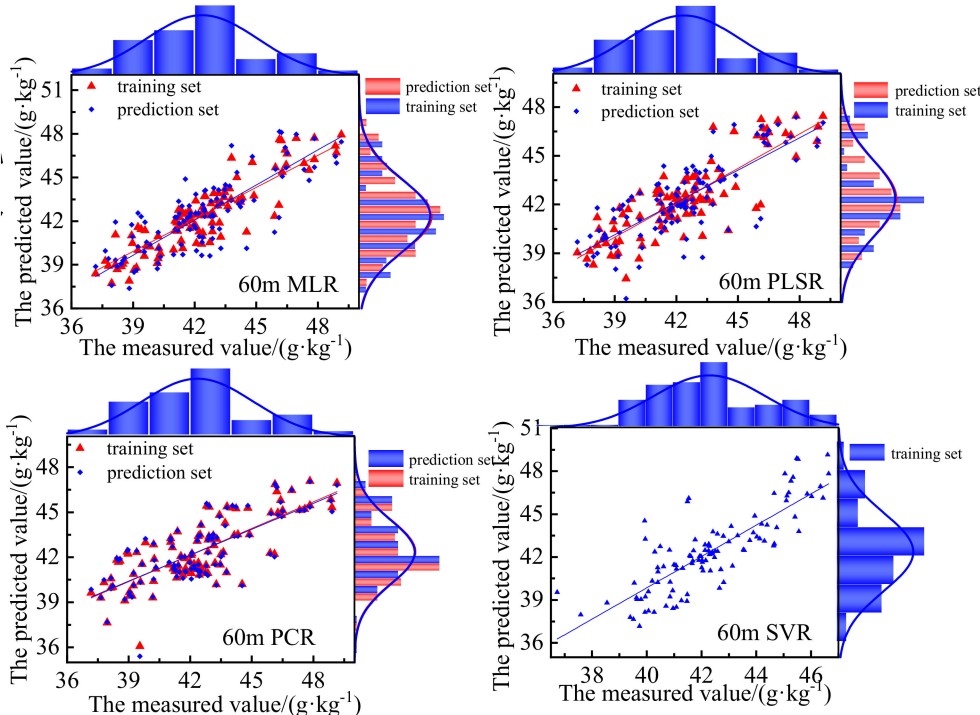

**Figure 14.** Scatterplots of the measured LNC vs. predicted LNC using multiple modeling methods (60 m).

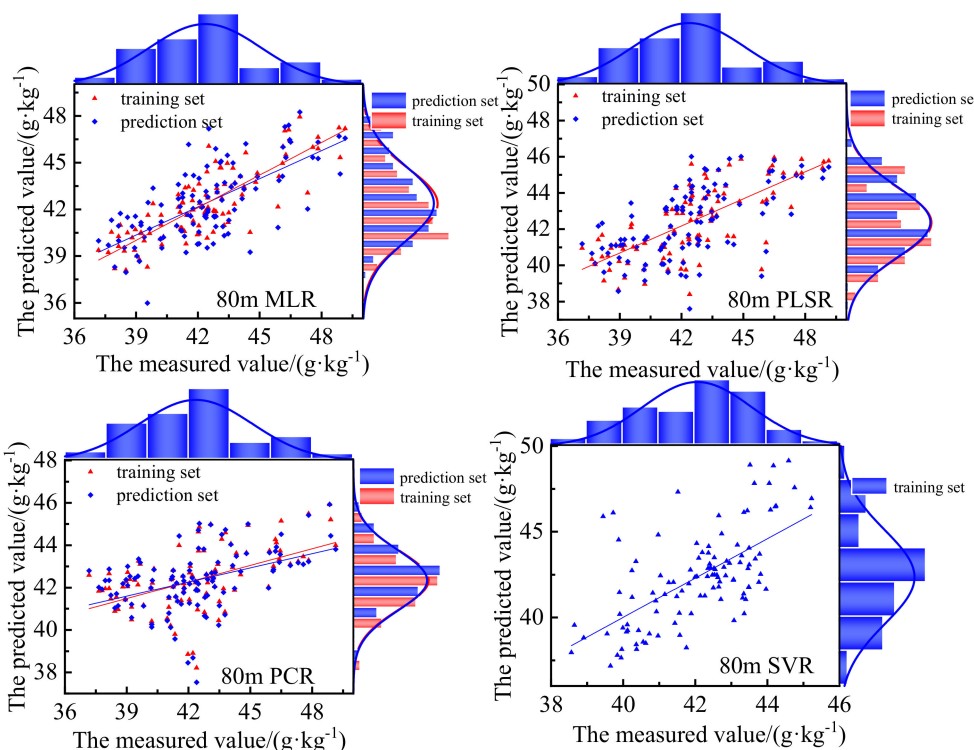

**Figure 15.** Scatterplots of the measured LNC vs. predicted LNC using multiple modeling methods (80 m).

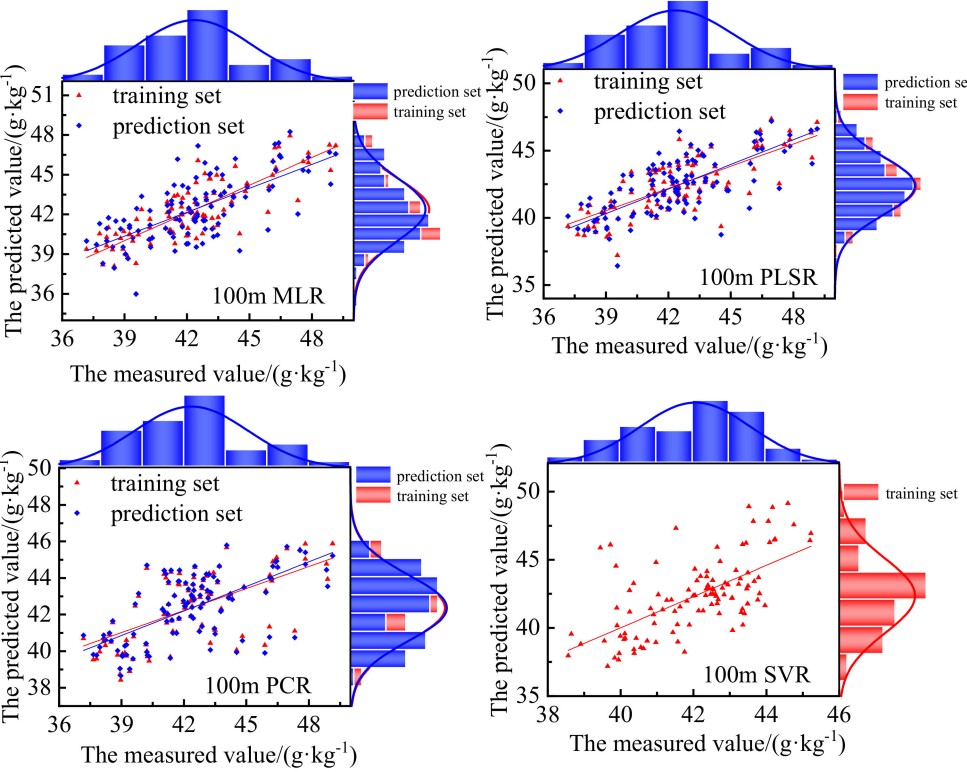

**Figure 16.** Scatterplots of the measured LNC vs. predicted LNC using multiple modeling methods (100 m).

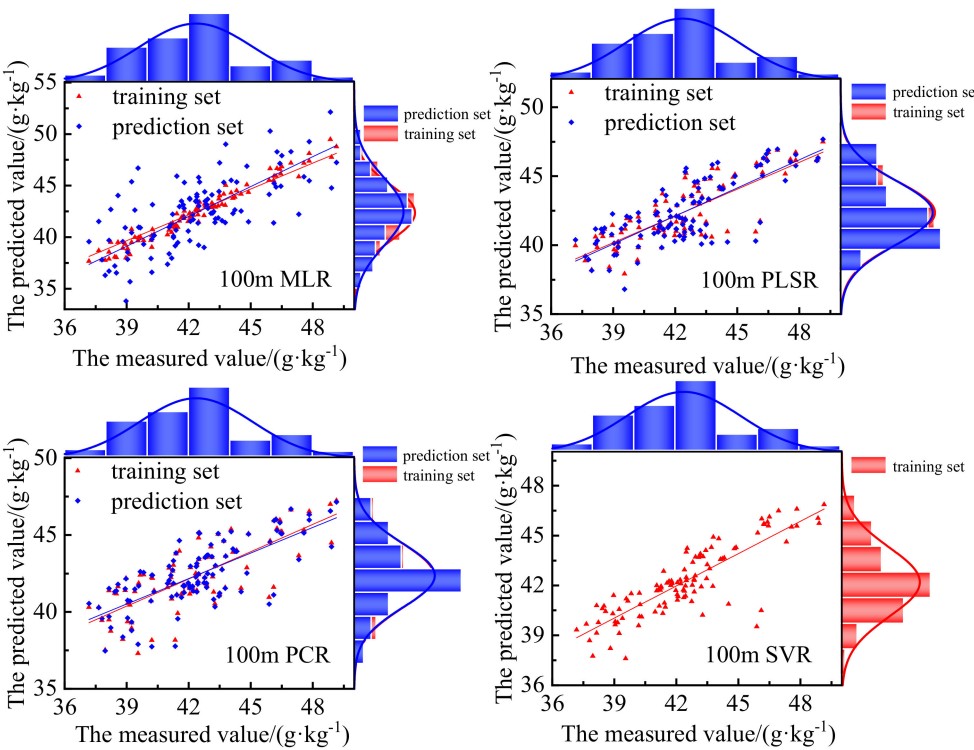

**Figure 17.** Scatterplots of the measured LNC vs. predicted LNC using multiple modeling methods (60, 80, and 100 m).

## 4. Discussion

According to the conclusions of Abulaiti et al. [48] with respect to vegetation spectroscopy of the Vis–NIR reflectance for chlorophyll concentrations, water content, and other substances in plants, obtaining the absorption coefficient of N from original spectra is difficult. Tian et al. pointed out that SR is positively correlated with LNC in visible light and negatively correlated with LNC in the near-infrared region [46]. This study shows that the correlation coefficient between the original spectrum of UAV and the LNC of cotton is very poor. However, the original reflectance obtained at a lower flight altitude (60 m) has a relatively high correlation with the nitrogen content of cotton leaves in both visible and near-infrared regions. SG and Gaussian filtering can significantly improve the correlation coefficient between SR and LNC. In this study, the correlation coefficient between SR before image classifying and LNC was not as high, and the correlation with LNC after classification was significantly enhanced. This is because the mixed pixels were less affected by the soil after classification. Abulaiti [48] proposed that the original spectra at 757 nm almost exhibited a positive correlation with total nitrogen content in cotton. In this study, the best indexes regarding the accurate determination of cotton LNC were $R_{800}/R_{550}$ (VI 3), $R_{NIR}/R_{Green}$ (VI 7), $(R_{780} - R_{550})/(R_{780} + R_{550})$ (VI 12), $(R_{810} - R_{560})/(R_{810} + R_{560})$ (VI 13), and $(R_{801} - R_{550})/(R_{801} + R_{550})$ (VI 22), and the correlation coefficients with cotton LNC reached more than 0.7. It could be seen that a common characteristic of these indexes was the SR at 550 nm (green light) and 800 nm. Caturegli [49] pointed out that the NDVIs obtained by the camera on a UAV are well correlated with the nitrogen content of turfgrass, and the bands used are 660 and 780 nm. Various scholars have concluded that the position of the response band of nitrogen content is in the visible and near-infrared regions, but that the specific bands are different. This may be caused by different crops or result from differences in the instrument band.

This study showed that with the increase in UAV height, the green peak decreases, the red valley increases, and the red edge position appears as a redshift. With the increase in spectral acquisition height, the obtained spectral curve mixes the information of soil [50].

The images acquired by the UAV within 60 m make it easier to monitor the LNC of cotton, but they are subject to many interference factors. Gaussian filtering and SG smoothing can be used to eliminate the redundant information from hyperspectral images [51]. In this study, the unsupervised classification of images obtained by a UAV was carried out, and it was found that the error between the classified spectral index samples was significantly reduced, which showed that mixed pixel decomposition reduces the interference of information from other sources such as soil [51]. At the same time, we found that a UAV image at 60 m can eliminate the interference of redundant information after Gaussian processing and SG smoothing. When the UAV is at 100 m, the SNV has similar functions.

In this study, the MLR was the most effective, followed by PLSR; SVR was slightly worse, and PCR was the worst. The predictive accuracy of PLSR models is thus inferior to SMLR models. This might be due to the latter selecting the most important indexes that affect the biochemical composition of plants by stepwise regression and then using the spectral reflectance of selected bands for estimating the biochemical composition of the plant [36]. In addition, some studies have demonstrated the optimal performance of the SMLR method in N estimations [52]. When data fusion modeling was conducted using data corresponding to three heights, the performance of the model was improved. In particular, the SVR modeling performance was now better compared with the other models. This shows that more abundant spectral information is obtained by a UAV flying at various altitudes, and the actual situation of a cotton canopy can thus be more comprehensively represented, a conclusion that is also supported by Raffy et al. [21].

Thus, these spectral indexes and corresponding monitoring models described in this article require further testing and improvements under a wider range of conditions. Future refinements will enable more accurate estimations with a potentially wide range of applications for field management systems. For example, in order for the UAV images to more comprehensively obtain cotton canopy information, we can increase the altitude and even the flight angle settings in future research. At the same time, multiple images can be captured on the same day, ensuring the image information is more representative.

## 5. Conclusions

There are significant differences in the SR of vegetation at different heights. With the increase in height, the green peak decreases, the red valley increases, and the red edge appears as a redshift. At the same time, the SR data obtained at heights of 60 and 80 m, 60 and 100 m, and 80 and 100 m are highly correlated, especially in the visible light range.

The unsupervised classification of UAV image pixels was carried out to extract the spectral information of the same category. After classification, the spectral error was significantly reduced. SG smoothing and Gaussian filtering can eliminate noise in spectral data. Pixel classification has a similar effect, which means that the obtained spectral data have few interference factors. When the UAV flies at a low altitude, the images can easily be influenced by a greater number of external factors. Gaussian smoothing and SG filtering are recommended for image preprocessing, but SNV is recommended for image preprocessing with the increase in UAV altitude.

In research relating to monitoring cotton LNC based on the UAV hyperspectral index, different modeling methods have different functions. In this study, the MLR had the best performance, followed by PLSR. SVR was slightly worse, and the performance of PCR was the worst. After data fusion modeling at three heights, the MLR showed an improved response to LNC. The performance of SVR from a single height was the worst, but the modeling effect was greatly improved following the fusion of data for three heights. It can be concluded that SVR has high requirements for model-independent variables.

**Author Contributions:** Conceptualization, C.Y. and X.L.; methodology, C.Y.; software, C.Y. and L.M.; validation, H.W.; formal analysis, L.Z. (Linshan Zhang); investigation, C.Y., L.M., and H.W.; resources, C.Y.; data curation, X.L.; writing—original draft preparation, C.Y.; writing—review and editing, Z.Z. and L.Z. (Linshan Zhang); visualization, Z.Z.; supervision, L.Z. (Lifu Zhang) and X.L.; project

administration, Z.Z.; funding acquisition, Z.Z. All authors have read and agreed to the published version of the manuscript.

**Funding:** This research was funded by the National Natural Science Foundation of China, grant number "42061658" and the Plan for Tackling Key Scientific and Technological Problems in Key Fields of Production and Construction Corps, grant number "2020AB005".

**Data Availability Statement:** Not applicable.

**Acknowledgments:** Thanks to my tutors and classmates for your selfless contributions. Thanks to everyone who helped me finish the experiment. Thanks to the journal editor and all experts for their practical suggestions. Because I met so many kind and selfless people along the way, all the efforts are worth it.

**Conflicts of Interest:** The authors declare no conflict of interest.

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
