# Peer review of "Hyperspectral UAV Images at Different Altitudes for Monitoring the Leaf Nitrogen Content in Cotton Crops"

_remotesensing, doi:10.3390/rs14112576_

Round 1
Reviewer 1 Report
The main issues identified in the previous review where addressed by the authors in this version of the document.
Author Response
Dear reviewer:
Thank you for your contribution to our manuscript.
You commented that the main issues identified in the previous review where addressed by the authors in this version of the document. We didn't find your other opinions in the new revised manuscript. So we further checked and modified the grammar of the paper. We have tried our best to revise all the inappropriate sentences we think. If there are any questions about the manuscript, please don't hesitate to contact us. We will try our best to modify it to improve the quality of the paper.
Please see the attachment for details. thanks!
Sincerely!
Yin caixia

Reviewer 2 Report
I found this paper very interesting where several technical aspects were nicely implemented and explained sufficiently. Undoubtedly, authors invested huge amount of time and have made a great effort to produce this high-quality of research which is clearly structured and the language used is largely appropriate. As final decision, I see that this manuscript in its form and level deserves to be accepted for publication in your highly respected journal.
Author Response
Dear reviewer:
Firstly, thank you for your affirmation of our achievements, which has added a lot of courage and confidence to us. Secondly, We are very thank you for your professional suggestions and comments on our manuscript, which has greatly improved the quality of our manuscript. Finally, we further checked and modified the grammar of the paper. We have tried our best to revise all the inappropriate sentences. If there are any questions about the manuscript, please don't hesitate to contact us. We will try our best to modify it to improve the quality of the paper.
Please see the attachment for details. thanks!
sincerely!
Yin caixia

This manuscript is a resubmission of an earlier submission. The following is a list of the peer review reports and author responses from that submission.
Round 1
Reviewer 1 Report
Dear authors,
the article has serious flaws and I stopped reviewing it in detail in the middle of reading. The article is written in an incoherent manner. Most parts cannot be understand or are at least hard to understand or misleading. The information given in the introduction is an enumration of what appears randomly selected information from articles. The major flaw is that the introduction and you aims / hypothesis are not matching. The overal aim is not clear and reader has to gues why you have done this study.
Your English is very poor. Most sentences are misleading and hardly to understand. Especially in results the graphs and tables presented are very small, with insufficient information to understand. Your text seems not to match with graphs. The text is hard to understand. Often speculative. The text MUST be revised by a native speaker.
The information given in Materials & Methods is not suffiencent to follow your results. Important information on the experiment is missing. Must be revised completly.
There is no discussion at all. You do not discuss your results in front of available studies. You do not cite a single reference!!! UAV and hyperspectral measurements are state of art these days, but you do not even mention papers on the same topic. I have added a few in the attached PDF.
L413-438. You could not expect from the reader to follow you here. This is just a lenghtly enumeration of unsorted results.
Results, discussion, and conclusions are not meaningful. in Some cases you present common knowledge like that on the "green peak". In conclusiosn you mentioned "plastic film" (L468). This is just mentioned once in the text (L323). Why? When it is in the conclusions it must have a meaning. But it is unclear which. The conclusion drawn in L455-46 are not supported by data. How were the 28 different VI selected. You must mention, if they already have been used (then add a reference) or if you have crate them. Then how? You will find more examples in the attached PDF
Overall, it appears that the manuscript is in a very first stage and by far not ready for publication.
Sorry, but your article must be rejected.

Reviewer 2 Report
Summary
This paper proposes a method to predict cotton leaf nitrogen content. Its main contributions consist in using hyperspectral data acquired with UAV at different altitudes over 2 cotton varieties with 6 nitrogen treatments.
Broad comments
The document is very hard to read and follow.
The English needs extensive review.
The document is well supported with references although the majority are very old.
The subject of the paper is interesting and with a great potential of application.
One of the biggest weaknesses of this study is the quality of the explanation of the procedures in the document. Frequently during reading the explanations are very general, with a lot of unneeded information or not well separated. Repetitive ideas with general conclusions.
The document needs extensive review, clearer ideas and explanations and better structured in order to be considered for publication.
Specific comments
In line 10 please correct “…and hyperspectral meter…”
In line 17 please rephrase “…The unsupervised classification to images pixels…”
Figure 1 is presented before it is referenced in the text. Please correct.
In Figure 1 the blue region is not identified in the green map. Please correct.
In line 130 and 131 it is not clear why the value 480 is located before value 330. Please explain and correct.
Table 1 is not formatted. Please correct.
Figure 3 does not exist. Please correct.
In line 153 please correct “…correction(figure.4)”
Detailed information in Table 2 and Table 3 is not necessary. General information can be included in the document text. Please correct.
In line 204 authors say that “…window points in this test is 11”. It is not clear from the text why this number. Please explain and correct in the text.
In line 219 please correct “…Filter model variables, The correlation”.
Equation 2 shows in the text before it is referenced in the text. Please correct.
Table 4 has no caption. Please correct.